# The Preliminary Chronic Effects of Electromagnetic Radiation from Mobile Phones on Heart Rate Variability, Cardiac Function, Blood Profiles, and Semen Quality in Healthy Dogs

**DOI:** 10.3390/vetsci9050201

**Published:** 2022-04-21

**Authors:** Van Nhut Khanh Dong, Lalida Tantisuwat, Piyathip Setthawong, Theerawat Tharasanit, Saikaew Sutayatram, Anusak Kijtawornrat

**Affiliations:** 1The International Graduate Program of Veterinary Science and Technology (VST), Faculty of Veterinary Science, Chulalongkorn University, Pathumwan, Bangkok 10330, Thailand; 6278501031@student.chula.ac.th; 2Department of Physiology, Faculty of Veterinary Science, Chulalongkorn University, Pathumwan, Bangkok 10330, Thailand; 6175525331@student.chula.ac.th; 3Department of Obstetrics, Gynaecology and Reproduction, Faculty of Veterinary Science, Chulalongkorn University, Pathumwan, Bangkok 10330, Thailand; piyathip.s@chula.ac.th (P.S.); theerawat.t@chula.ac.th (T.T.); 4Veterinary Clinical Stem Cells and Bioengineering Research Unit, Chulalongkorn University, Pathumwan, Bangkok 10330, Thailand

**Keywords:** cardiac troponin I, dog, echocardiography, electrocardiography, electromagnetic radiation, heart rate variability, mobile phone, semen quality

## Abstract

The present study aims to determine the effects of long-term exposure to electromagnetic radiation from mobile phones (MPs) on heart rate variability (HRV), cardiac function, blood profiles, body surface temperature, and semen quality in healthy dogs. Eight male dogs were exposed to MPs (1962–1966 MHz; specific absorption rate 0.96 W/kg) for 2 h/day, 5 days/week, for 10 weeks. Holter monitoring for HRV analysis was performed at baseline (BL) and every 2 weeks, until the end of the study. Electrocardiograms (ECG), blood pressure (BP), echocardiography, cardiac troponin I (cTnI), hematology and biochemistry profiles, body surface temperature, and semen quality were evaluated at BL, week 5, and week 10 during exposure. The results showed that most of the HRV parameters did not significantly differ among timepoints, except for the mean of an interval between continuous normal R waves in week 6 that was higher than that at BL (*p* = 0.022). The RR and QT intervals from ECG in week 5 were prolonged, compared to the BL values (*p* = 0.001 and *p* = 0.003, respectively), but those parameters were within the normal limits. The echocardiography, BP, cTnI concentrations, body surface temperature, and semen quality results were not different from BL values. In conclusion, this study found no evidence suggesting an adverse effect of cell phone exposure on HRV, cardiac function, blood profiles, body surface temperature, or semen quality in healthy dogs, when exposed for 10 weeks.

## 1. Introduction

Mobile phone (MP) usage has been increasing enormously worldwide, leading to widespread electromagnetic radiation (EMR) in the living environment. Unlike X-rays or gamma rays, EMR from MPs is classified as non-ionizing radiation that can neither break chemical bonds nor cause ionization in humans [1]. However, long-term exposure to EMR, emitted from MPs, may potentially cause health hazards for both humans and animals, since body tissues could absorb this radiation [2,3]. Previous studies showed that long-term exposure to EMR could induce both thermal or non-thermal effects [4,5]. Moreover, EMR could cause oxidative stress injury in the brain [6,7] and heart tissues [8,9,10] in rat models. In other studies, functional alterations in the central nervous system (CNS), cardiovascular system (CVS), and hemopoietic system were also reported after long-term exposure to EMR from MPs in animals [6,8,11]. Nevertheless, the pathophysiological mechanisms of the EMR from MPs have not been fully clarified. Therefore, this issue has raised public concern and attracted recent research interest.

The autonomic nervous system (ANS) is the major controller of the respiratory, cardiovascular, digestive system, endocrine, and other systems. The ANS controls the heart rate (HR) via the parasympathetic nervous system (PNS) and sympathetic nervous system (SNS). Therefore, heart rate variability (HRV), an analysis of the oscillation of the inter-beat interval in electrocardiography (ECG) tracing, is considered to be a non-invasive, indirect method that can be used as an index of ANS modulation to the heart [12,13]. In humans, several studies have shown that HRV was altered during acute exposure to EMR from MPs, such as an increase in parasympathetic tone, concurrent with a decrease in sympathetic tone [14,15]. Other studies on the chronic effects of MPs in healthy humans have reported a decrease in parasympathetic tone and increase in sympathetic tone [16,17].

Regarding the CVS, long-term exposure to EMR could have altered the CVS function in a rat model [8]. In addition, previous studies suggested that long-term exposure to EMR generated free radicals, such as reactive oxygen species (ROS), leading to oxidative stress injury [8,9,10] and cardiomyocyte structural alterations in rats [8,9,18]. In clinical applications, the most common methods used to evaluate cardiac electrical, mechanical, and hemodynamic functions are ECG, echocardiography, and blood pressure (BP) measurements, respectively. Additionally, cardiac troponin I (cTnI)—a sensitive and specific biomarker—has been used widely to detect myocardial injury [19]. Numerous studies have attempted to verify the influences of EMR on cardiac electrical activity and hemodynamic function in both human subjects and rodent models [8,20,21,22]. However, it has not yet been determined whether the EMR from MPs affects cardiac mechanical function and cardiomyocyte damage using echocardiography and cardiac biomarkers in dog models.

In addition to the ANS and CVS, blood profiles are the most crucial criteria for health status evaluations in both humans and animals. Previous studies showed that the effects of EMR on the hematological and blood biochemical profiles are still discrepant in both animals and humans [23,24,25,26]. In addition, EMR might alter reproductive endocrine mechanisms, especially in males, since males usually carry MPs in their pant pockets, which may affect testicular functions. A previous study in rabbits demonstrated that exposure to EMR, from MPs for 8 h/day for 8 weeks, led to a significant decline in the sperm count, and sperm motility was decreased after exposure for 10 weeks [27]. Moreover, rats exposed to MP emissions (6 h/day) for 18 weeks exhibited a significantly higher incidence of sperm cell death than the control group [28]. In contrast, no evidence of negative effects of MP exposure on testicular function and structure was observed in several studies [29,30,31]. The uncertainty of such results is attributed to many factors, such as the study design, source of EMR, frequency range (e.g., 800, 848.5, 900, and 915 MHz), and animal species. Most studies related to the effects of EMR from MPs on male reproductive system were based on experimentation with rodents and rabbits. Studies of EMR effects in non-rodent species are necessary, in order to fully understand these findings.

Currently, the effects of long-term exposure to EMR emitted from MPs on the ANS’s control of the heart, CVS function, blood profiles, and semen quality in healthy dogs have not been investigated simultaneously. Therefore, we aimed to determine the effects of chronic exposure to EMR emitted from MPs on HRV, ECG, echocardiography, BP, cTnI, hematology and blood chemistry profile, and semen quality in conscious dogs.

## 2. Materials and Methods

The study was conducted at the Chulalongkorn University Laboratory Animal Center (CULAC), Thailand, and approved by the Institutional Animal Care and Use Committee of CULAC (animal use protocol number: 2073013). All of the procedures of this experiment followed the rules for the care and use of laboratory animals, in compliance with the Animals for Scientific Purposes Act (2015 A.D.) [32].

### 2.1. Animals

Eight male beagles (*Canis familiaris*), aged 1 year old and weighing 11–15 kg, were enrolled in the study. Dogs were acclimatized in appropriate dog runs for one week before the experiment. Health screenings including a complete physical examination, blood profile analysis, electrocardiography (ECG), echocardiography, and blood pressure measurements, performed to exclude systemic diseases. The animal room was maintained at the temperature of 22 °C ± 1 °C, with a relative humidity of 50% ± 20%, and a dark:light cycle of 12 h:12 h was used. Dogs were fed commercial dog food once daily at 11:30 a.m. and able to access water freely.

### 2.2. EMR Exposure

The EMR was produced using a Samsung GT-E3309I MP (Samsung Electronics Co., Ltd., Gyeonggi, Korea) with an advanced info service (AIS) signal (1962–1966 MHz). According to the manufacturer’s information, this MP model created a specific absorption rate (SAR) at 0.96 W/kg, when measured at a distance of approximately 1.5 cm from the body [33]. Two MPs were placed on each dog’s chest (i.e., one MP for each side) and secured inside the pockets of a customized jacket. The MPs were set on silent and non-vibrate modes, and the calls were performed from 8:00 a.m. to 5:00 p.m. to avoid physical stimulation and circadian effects. The MPs were dialed from another phone, located at a different building, to create a total incoming call duration of 2 h daily (i.e., 120 calls of 1 min of ringing, with 3.5 min between calls), 5 days a week, for 10 weeks.

### 2.3. Experimental Designs

All dogs were subjected to semen collection training for 10 consecutive weeks before the beginning of the main experiment. After that, the main experiment began with baseline measurement of 24 h Holter monitoring, limb lead ECG recording, echocardiography evaluation, BP measurement, semen collection, and blood collection for hematology, biochemistry analysis, and cTnI measurement before their exposure to EMR from MPs. After the beginning of the MP exposure, Holter monitoring for HRV analysis was performed every 2 weeks, until the end of the study. The remaining parameters were evaluated at 5 and 10 weeks after exposure. The body surface temperature at the location of MP placement was measured at the beginning and the end of the MP call (8:00 a.m. and 5:00 p.m., respectively) on the first day (D1), then every 5 weeks, until the end of the study.

### 2.4. Experimental Procedure and Analysis

#### 2.4.1. Ambulatory Holter Monitoring

ECG electrodes were attached to the thoraxes of dogs, using transthoracic leads, and connected to the Holter monitor device (Fukuda Denshi Co., Ltd., Tokyo, Japan), as previously described [34]. Dogs were subjected to continuous ECG recording for 24 h in individual dog runs without restraints. After 24 h of recording, ECG data were analyzed using SCM-510 Holter software (Fukuda Denshi Co., Ltd., Tokyo, Japan), as previously described [35]. Briefly, all of the R-waves were detected and labeled automatically by the software. Then, the QRS complexes from ECG tracing were manually edited to identify and exclude the ectopic beats. The ECG recording was accepted only when R-waves were normal, up to 80%. The signals were filtered by means of the Hamming window, and then transformed into frequency spectrums using fast Fourier transformation. HRV parameters were analyzed at the consecutive RR interval of 512 samples. The time- and frequency-domain parameters of HRV were evaluated according to the method recommended by the Task Force of the European Society of Cardiology [12]. The time-domain parameters were the mean of an interval between the continuous normal R waves (NNA), standard deviation of all NN-intervals (SDNN), standard deviation of the averages of NN-intervals in all 5 min segments of the entire recording (SDANN), mean of the standard deviation of the averages of NN-intervals in all 5 min segments of the entire recording (SDNN index), root mean square of successful differences from each other between adjacent NN-intervals (rMSSD), and percentage of adjacent NN-intervals differing by more than 50 milliseconds (ms) in the entire recording (pNN50). The frequency bands used in this study were the ultralow frequency—ULF (≤0.004 Hz), very low frequency—VLF (0.004–0.041 Hz), low frequency—LF (0.041–0.15 Hz), high frequency—HF (0.15–0.50 Hz), LF-to-HF ratio, normalized LF power—LF nu (LF nu = LF/LF + HF), and normalized HF power—HF nu (HF nu = HF/LF + HF) [36].

#### 2.4.2. Electrocardiography Recording

The limb lead ECG recording was obtained using an ECG machine (CardiMax FX–7102, Fukuda Denshi Co., Ltd., Tokyo, Japan). The dog was positioned in right lateral recumbency, and the four ECG clips were attached to dog’s four limbs to record the ECG, including bipolar limb (i.e., lead I, II, and III) and unipolar limb (i.e., aVR, aVL, and aVF) leads. The ECG parameters, comprising of the RR interval, PQ interval, QRS duration, and QT interval, were measured manually using Vernier calipers, as previously described [37]. An average of 6 consecutive cardiac cycles was calculated and presented as the result. The corrected QT (QTc) was calculated using Van de Water’s formula (QTc = QT − 0.087 × (RR − 1000)) [38].

#### 2.4.3. Echocardiography

Echocardiography was performed on conscious dogs using an echocardiographic system (M9, Mindray Bio-Medical Electronics Co., Ltd., Shenzhen, China) by an experienced veterinarian, as previously described [39]. The 2-D, M-mode, and pulsed-wave Doppler echocardiography processes with continuous ECG recording were performed using a 2–4 MHz phased array cardiac probe and following the recommendations of the Echocardiography Committee of the Specialty of Cardiology of the American College of Veterinary Internal Medicine [40]. Two-dimensional echocardiography was performed in the right parasternal short- and long-axis views to assess the cardiac structure. M-mode echocardiography was carried out during diastole and systole on the right parasternal short-axis view to evaluate the cardiac chamber diameters, the myocardial wall thickness, and the systolic function. Pulse-wave Doppler echocardiography was performed to measure the maximum of the aortic valve pressure gradient (AV PG max) from the left apical five-chamber view [41]. The normalized dimensions of the left ventricular internal dimension diastole (LVIDd) and left ventricular internal dimension systole (LVIDs) were calculated based on the animal body weight (BW), according to the formula: normalized LVIDd (LVIDDN) = LVIDd/BW^0.294^ and normalized LVIDs (LVIDSN) = LVIDs/BW^0.315^ [42]. The modified Simpson method was used to measure the cardiac output (CO) on the right parasternal long-axis four-chamber view. The Tei index was obtained to evaluate both systolic and diastolic functions using dual-phased Doppler. The Tei index was manually calculated, based on the isovolumetric contraction time (IVCT), isovolumetric relaxation time (IVRT), and left ventricular ejection time (LVET), following the formula: Tei index = (IVCT + IVRT)/LVET [43].

#### 2.4.4. Blood Pressure Measurement

The systolic, diastolic, and mean arterial blood pressure were measured using an oscillometric device (Pet MAP^TM^ graphic II system, Ramsey Medical, Inc., Tampa, FL, USA), as previously described [35]. In brief, the dog was placed in the right lateral recumbency, and BP was measured at the median artery of the left front leg. The selection of cuff sizes and the acceptable reading curve values were based on the company’s recommendation. Three consecutive measurements of BP with a good reading quality were recorded and averaged.

#### 2.4.5. Hematology and Biochemistry Profiles and Cardiac Biomarker (cTnI) Analysis

A total of 3 mL of whole blood was collected from the cephalic vein. Half a milliliter of blood was collected in an ethylene diamine tetra-acetic acid (EDTA) tube for automated complete blood count (CBC) analysis (Mindray BC-5000VET, Shenzhen Mindray Bio-Medical Electronics Co., Ltd., Shenzhen, China). Another 0.5 mL of blood was collected in a lithium heparinized tube for automated biochemistry analysis (Mindray BS-800, Shenzhen Mindray Bio-Medical Electronics Co., Ltd., Shenzhen, China). The remaining blood was collected in a lithium heparinized tube; then, the plasma was collected, after centrifuging at 3000 rpm for 10 min. The plasma samples were stored at −80 °C for further analysis of the cTnI level. The concentrations of cTnI were measured with the Architect STAT highly sensitive troponin I immunoassay using an Alinity™ ci-series analyzer (Abbott Laboratories, Abbott Park, IL, USA) with a detection range of 0.001–50 ng/mL.

#### 2.4.6. Body Surface Temperature Measurement

The body surface temperature on the chest, where the MP located, was measured at the beginning (8:00 a.m.) and end of the MP call period (5:00 p.m.) on the first day (D1) of MP call exposure, then every 5 weeks during the study, using a body infrared thermometer (YMITF01, Tecpel Co., Ltd., New Taipei City, Taiwan, China).

#### 2.4.7. Semen Collection and Analysis

All dogs were trained for routine semen collection once per week for 10 consecutive weeks before the beginning of the experiment. Then, the baseline data were collected. After that, dogs were exposed to EMR from MPs, as described in the previous section. During the experiment, semen was collected once a week, and the semen collections in weeks 5 and 10 were used for further analysis, whereas the rest were discarded. Each semen sample collected at baseline, and the semen at 5 and 10 weeks was processed for the determination of volume, sperm motility, sperm concentration, morphology, sperm viability, and DNA integrity, as previously described [44]. Briefly, the semen ejaculation was collected into 15 mL Falcon conical tubes via digital manipulation with a latex glove. The method was performed in all dogs, without the presence of an estrous bitch. Basically, the dog’s penis was vigorously massaged until the presence of bulbus glandis engorgement. Then, the collector retracts the prepuce caudally past the bulbus glandis and applies firm constant pressure to the penis behind the bulbus glandis using index finger and thumb. The ejaculation is usually started after the application of pressure behind the bulbus glandis. The total duration of induced ejaculation was 5 min per dog. The total semen volume was evaluated from three consecutive fractions, which included the pre-sperm fraction, sperm-rich fraction, and prostatic fraction. Semen samples were kept at 37 °C, until analysis.

The volume of the sample was determined using a graduate micropipette (Gilson, Gilson International France, Villiers-le-Bel, France).

The sperm concentration was determined using a hemocytometer. A raw semen was placed in a channel for sperm counting and covered with its special coverslip to allow a very specific amount of fluid to be contained under the coverslip. The total sperm concentration was counted at 400× magnification, using a phase contrast microscope (Olympus, L’Hospitalet de Llobregat, Spain).

Morphological assessment of the sperm tail and head was performed using wet-mounting of formal saline-fixed samples and Williams staining, respectively. A total of 200 sperm cells were evaluated at 400× magnification. The percentages of normal head and tail morphologies were reported.

The percentage of sperm motility was evaluated immediately after collection. A 10 μL sperm sample was placed on a pre-warmed slide (37 °C) and covered with a pre-warmed coverslip. The motility was evaluated as total motility (i.e., the number of sperm moving). Initially, it was assessed at low power, and the final observations were made at 100× magnification using a phase contrast microscope (Olympus, L’Hospitalet de Llobregat, Spain).

Sperm viability was analyzed by mixing semen with Calcein AM and Eth-idium homodimer-1 (EthD-1). The samples were incubated at 37 °C for 15 min. A total of 200 sperm cells were evaluated for viability under a fluorescent micro-scope at 400× magnification (Olympus, L’Hospitalet de Llobregat, Spain) and reported as a percentage of sperm viability.

DNA integrity was evaluated via staining with acridine orange. The normal DNA content showed green fluorescence staining. Sperm displaying yellow-orange to red fluorescence were considered to have damaged sperm DNA. A total of 200 spermatozoa were evaluated under a fluorescent microscope at 1000× magnification (Olympus, L’Hospitalet de Llobregat, Spain) and reported as a percentage of sperm with DNA integrity.

### 2.5. Statistical Analysis

All statistical analyses were performed using SAS version 9.4 (SAS Institute Inc., Cary, NC, USA). The Shapiro–Wilk test was used to determine the normal distribution. The HRV parameters, including the time-and frequency-domain components, were compared among the timepoints (BL, week 2, week 4, week 6, week 8, and week 10) using repeated one-way Analysis of variance (ANOVA) with Dunnett’s post hoc analysis or Friedman’s test for non-normally distributed data. The ECG, echocardiography, BP, cTnI concentrations, hematological and biochemical profiles, body surface temperature, and semen quality variables were compared among the timepoints (BL, week 5, and week 10) using repeated one-way ANOVA with Dunnett’s post hoc analysis or Friedman’s test for non-normally distributed data. Data were presented as mean ± standard deviation (SD). A *p*-value < 0.05 was considered statistically significant.

## 3. Results

One dog was excluded from the current study, since ectopic beats (i.e., ventricular premature complex) were detected at the BL timepoint via Holter monitoring. Physical examination results and other parameters of the remaining seven dogs, before beginning of the experiment, were within normal limits.

### 3.1. Heart Rate Variability (HRV)

The parameters of time- and frequency-domain components of HRV analysis are shown in Table 1. Most of the time-domain parameters (i.e., SDNN, SDANN, SDNN index, rMSSD, and pNN50) were not statistically different, when compared with BL. However, the NNA value obtained at week 6 was significantly higher than that of the BL timepoint (*p* = 0.022). The frequency-domain parameters showed no statistical difference, when compared with the BL values.

### 3.2. Cardiac Function and Biomarkers

#### 3.2.1. Electrocardiography

The ECG parameters are presented in Table 2. The PQ interval, QRS duration, and QTc values showed no statistically significant differences, in comparison to BL values. However, the RR interval and QT interval were significantly prolonged in week 5, compared to BL (*p* = 0.001 and *p* = 0.003, respectively).

#### 3.2.2. Echocardiography

The echocardiographic data are summarized in Table 3. All of the echocardiographic parameters obtained at weeks 5 and 10 did not achieve statistical significance, compared with BL values.

#### 3.2.3. Blood Pressure Measurement

The hemodynamic parameters, including the average systolic blood pressure (SBP), diastolic blood pressure (DBP), and mean arterial blood pressure (MABP), at the BL timepoint were 162.5 ± 16.6 mmHg, 90.1 ± 16.2 mmHg, and 113.5 ± 15.7 mmHg, respectively, all of which were not statistically different, compared to the week 5 timepoint (SBP 163.3 ± 8.5 mmHg, DBP 93.3 ± 10.8 mmHg, and MABP 118.1 ± 9.5 mmHg) and week 10 timepoint (SBP 165.9 ± 16.5 mmHg, DBP 96.5 ± 8.5 mmHg, and MABP 120.1 ± 11.4 mmHg).

#### 3.2.4. Cardiac Biomarker (cTnI)

There was no significant difference in the cardiac troponin I concentrations at the BL timepoint (0.0025 ± 0.0020 ng/mL), compared to those of week 5 (0.0025 ± 0.0007 ng/mL) and week 10 (0.0030 ± 0.0008 ng/mL).

### 3.3. Complete Blood Count (CBC) and Plasma Biochemistry Profiles

The hematological and plasma biochemical profiles are shown in Table 4. There was no statistically significant difference in the red blood cell (RBC), hemoglobin, or hematocrit values, compared to the BL timepoint. Regarding the RBC indices, only the mean corpuscular hemoglobin concentration (MCHC) values at the week 10 timepoint were significantly lower (*p* = 0.025) than the BL values. The platelet, lymphocyte, and monocyte concentrations were not different from BL. Additionally, the white blood cell (WBC), neutrophil, eosinophil, and basophil concentrations at week 10 were significantly lower than those at BL (*p* < 0.001).

Regarding the biochemical profiles, the alanine transaminase (ALT) and alkaline phosphate (ALP) concentrations did not differ from BL values, whereas the aspartate aminotransferase (AST) concentrations were increased in both the week 5 and 10 timepoints (*p* = 0.002). The creatinine concentrations were similar throughout the study timeline, whereas the blood urea nitrogen (BUN) concentrations increased in both the week 5 and 10 timepoints, compared with the BL values (*p* < 0.001 and *p* = 0.008, respectively). The total protein concentrations at weeks 5 and 10 were higher, compared with those at BL (*p* = 0.031 and *p* < 0.001, respectively). An increase in albumin concentrations was found only at week 10, compared to BL (*p* < 0.001).

### 3.4. Body Surface Temperature

The temperature in the morning (8:00 a.m.) and afternoon (5:00 p.m.), as well as the changes in temperature throughout the day (∆ temperature), are shown in Table 5. There was no significant difference in the body surface temperature at both weeks 5 and 10 timepoints, compared to BL.

### 3.5. Semen Quality

#### 3.5.1. Semen Volume

The average volume of semen samples collected from all seven dogs was 4.65 ± 3.62 mL at BL, 4.77 ± 2.30 mL at week 5, and 4.79 ± 2.84 mL at week 10. There were no significant differences among timepoints.

#### 3.5.2. Total Sperm Count

The total sperm counts from the collected semen did not statistically significantly differ among timepoints, as determined through one-way ANOVA (*p* > 0.05). At BL, we observed a mean of 467 ± 276 × 10^6^ cells/mL; at 5 and 10 weeks of exposure to EMR, we observed a mean of 453 ± 297 × 10^6^ cells/mL and 487 ± 225 × 10^6^ cells/mL, respectively.

#### 3.5.3. Sperm Morphology

The average percentages of normal sperm heads vs. tails in the semen samples collected at BL, week 5, and week 10 were 96.6% ± 2.07% vs. 95.7% ± 2.65%, 97.1% ± 1.27% vs. 95.1% ± 4.13%, and 97.2% ± 1.71% vs. 97.2% ± 0.62%, respectively. There were no significant differences among timepoints for each parameter.

#### 3.5.4. Sperm Motility and Viability and DNA Integrity

The average sperm motility and viability of semen samples collected from all seven dogs were 83.0% ± 2.7% and 80.2% ± 3.8% at BL, 80.0% ± 4.5% and 81.1% ± 3.2% at week 5, and 85.0% ± 7.1% and 80.3% ± 4.0% at week 10, respectively. The average DNA integrity of semen samples was 99.4% ± 1.7%, 99.8% ± 0.2%, and 98.6% ± 1.1% at BL, 5 weeks, and 10 weeks, respectively. There were no significant differences among timepoints for each parameter.

## 4. Discussion

In the present study, we assessed the biological effects of long-term MP exposure at a general SAR on the HRV, CVS function, blood profiles, body surface temperature, and semen quality in healthy dogs. Although we detected statistically significant differences of some variables between exposure timepoints and BL, none of those differences were clinically significant when the dogs were expososed to EMR for 10 weeks.

### 4.1. Heart Rate Variability

This study demonstrated that chronic exposure to EMR, emitted from MPs (1962–1966 MHz) for 10 weeks, did not significantly alter most of the HRV parameters. However, only the NNA value at week 6 statistically increased, when compared to the BL value. The NNA is the mean of the normal RR interval lengths, which is reciprocal to HR, and it can be changed through an alteration in ANS function [12]. In addition, other factors may also affect the HR and HRV results, such as respiration, emotions, posture, mood, and hormones (i.e., thyroid, sexual hormones, and cortisol) [45]. Higher NNA values (i.e., lower HR) may suggest an increase in vagal activity. However, other HRV parameters, related to augmented vagal activity for both time-domain (i.e., increased rMSSD and pNN50) and frequency-domain (i.e., increased HF and HFnu, and/or decreased LF/HF and LFnu) parameters, did not show a significant change at week 6. Therefore, the increase in NNA in the current study might be a random variation. Interestingly, previous studies in healthy humans revealed a decrease in parasympathetic tone and increase in sympathetic tone, when exposed to chronic MP usage [16,17]. Moreover, another chronic study in a rat model showed that EMR exposure from MPs (900/1800 MHz) for 6 weeks (10 min/day) reduced the HRV (i.e., increased LF components) [46]. In a study of acute EMR exposure in humans, 15 min of EMR, emitted during an MP call (1800 MHz), increased the parasympathetic tone (inferred from an increased HF) and concurrently with a reduction in sympathetic tone (inferred from the reduction in LF/HF), compared to before and after the MP call. Another study in anesthetized rabbits demonstrated that 150 min of EMR exposure, using an 1800 MHz EMR device generator or the base stations of mobile providers, increased HF and rMSSD values, which related to the enhancement of vagal control [47]. Therefore, the significantly increased NNA observed in this study may require further investigation to confirm whether or not this fluctuation in NNA was caused by an alteration in vagal activity.

### 4.2. Cardiac Function and Biomarkers

In the present study, long-term exposure to EMR did not clinically affect cardiac function or cause cardiac injury in healthy conscious dogs, as inferred based on most of the variables obtained in the study. A previous study in rats, with 8 weeks of exposure to MPs (2–3 h/day), demonstrated a significant increase in the ECG variables (i.e., PR, QTc interval, and QRS duration), which was accompanied by the augmentation of renin activity and left ventricular hypertrophy with myocardial hypertrophy and alterations. In addition, plasma calcium concentrations were significantly lower in EMR exposure groups, for which the author proposed the interaction mechanisms between EMR and intracellular ions [8]. In the present study, most of the ECG parameters did not differ from BL values, with the exception of the RR and QT intervals, which were significantly prolonged at week 5, compared to the BL values. We do not know whether this change related to the incidentally increased NNA values of HRV observed in week 6. Although RR and QT intervals at week 5 were statistically prolonged, compared with BL values, other ECG variables and all of the cardiac function parameters and cTnI were within normal limits. We are aware that other factors, such as the dogs becoming familiar with the experimental procedures and veterinarians, leading to slightly lowering augmentations in sympathetic tone and enhancement of the vagal tone, could occur coincidently and might mask the real effects of EMR found in the current study.

Furthermore, our results showed that 10 weeks (2 h/day) of exposure to MPs did not significantly influence the cardiac mechanical function and structure inferred from variables obtained via echocardiography and cTnI. In addition to the stable cardiac injury biomarkers found in this study, cTnI values at all timepoints were quite low, compared to the normal reference range of dogs [48,49,50]. Our results conflict with several studies in rat models, which have postulated that long-term exposure to EMR leads to increased oxidative stress that could damage the cardiac tissue and, ultimately, alters the cardiac structure and function [8,9].

In the present study, 10 weeks of exposure to EMR did not affect BP levels. Similarly, Tahvanainen and colleagues (2004) found that short-term exposure to MP (35 min) did not alter and hemodynamic function in a human study [51]. Likewise, chronic EMR (1800 MHz) exposure in a rat model, for a total of 3 months (2 h daily), did not create a significant difference in BP parameters, compared with the sham group [52]. However, our result is in disagreement with another chronic study in rats, in which increased SBP was found after 8 weeks of (2–3 h/day) EMR exposure from MPs. These rats also developed an increase in renin activity, hypocalcemia, and left ventricular hypertrophy [8]. Additionally, 6 weeks (10 min/day) of EMR exposure from MPs (900/1800 MHz) altered the BP in another rat model (i.e., increased in SBP, DBP, and MABP) [46]. This difference may be explained by the difference in the body size of the subjects, as supported by several studies. A previous study that compared the SAR in the adults and children showed that children with smaller heads and thinner skulls tended to absorb more energy, compared to adults [2]. In addition, the heart is protected by several body tissues (e.g., the ribcage, muscle layer, and lungs), and it is located at some depth from the location of MP placement. Therefore, subjects with a large body size or deeper organs might show effective resistance to the EMR emitted from MPs, as a thicker barrier between the EMR source and affected tissues could reduce the EMR intensity for the target tissues.

### 4.3. Hematology and Biochemistry

In the current study, there was a minor change in blood indices (i.e., MCHC) within the normal range, whereas the RBC, hemoglobin, and hematocrit were not significantly altered from the BL values. Our results were similar to a previous rat study, in which RBC, hemoglobin, and hematocrit values showed no significant change after exposing the rats to 900 MHz and 1800 MHz of EMR for 3 months (2 h/day) [52]. Previous studies of long-term EMR exposure in rodent models demonstrated a significant decrease in hematocrit, RBC, and hemoglobin values [53,54]. This reduction could be explained by susceptible oxidative damage induced by EMR of the hematopoietic system, such as the spleen tissues, thymus, and bone marrow [55], as well as cytotoxic and genomic effects of EMR on bone marrow cells [56].

Although the WBC and neutrophil concentrations significantly decreased in the current study, after the dogs were exposed to EMR, those changes were still within the normal range. In this study, the lymphocyte and monocyte concentrations were unchanged, whereas the eosinophil and basophil concentrations decreased significantly after 10 weeks of MP exposure. These WBC alterations were inconsistent in a rat study [52], and the authors barely mentioned the pathophysiological mechanisms for this. Currently, the results on the effects of EMR exposure on hematology are still varied, depending on the EMR duration and power intensity, as well as the animal models, all of which could interfere with the EMR effects on blood parameters.

Regarding blood chemistry profiles, 10 weeks of exposure to EMR in this study induced a slight increase in the AST, BUN, total protein, and albumin concentrations, whereas the creatinine concentration did not change. Increased AST concentrations and some liver proteins may indicate liver damage; however, other liver enzymes (i.e., ALT and ALP) were not significantly elevated, and the AST alterations were also within the normal range in this study. Therefore, the slight increase in some liver parameters was likely to be a coincidence, rather than a clinical effects of EMR on liver function.

### 4.4. Body Surface Temperature

In our study, similarly, body surface temperature was not clinically elevated, compared with the BL timepoint. Although changes in temperature within day were found in this study, the increase in body surface temperature under the dog jacket could be a result of wearing the dog jacket and circadian effects (i.e., higher levels of physical activity and metabolism during the daytime). Therefore, the thermal effects of EMR exposure might not be clinically significant in these healthy dogs.

### 4.5. Semen Quality

No significant differences in semen quality—semen volume, total sperm counts, sperm morphology, motility and viability, and DNA integrity—were detected in dogs exposed to 10 weeks of EMR at 1962–1966 MHz with an SAR of 0.96 W/kg. The results of this study, related to semen quality, are consistent with the previous findings of several investigators [57,58]. Ozguner and colleagues (2005) showed that exposure of Sprague–Dawley (SD) rats to 30 min/day, 5 days/week, for 4 weeks to 900 MHz electromagnetic field emitted from electromagnetic energy generator (2 watts peak power, average power density 1 mW/cm^2^), did not show adverse effect on spermatogenesis or on germinal epithelium [57]. In addition, Dasdag and colleagues (2008) demonstrated that Wistar rats exposed to 900 MHz radiation (2 W/kg, 2 h/day, 7 days/week, for 10 months) did not exhibit effects on the active caspase-3 levels in testes [58]. Furthermore, Lee and colleagues (2010) showed that exposure of male SD rats to radiofrequencies of 848.5 MHz at 2 W/kg for 12 weeks showed no alterations in testicular functions, in terms of the cauda epididymis sperm counts, frequency of spermatogenesis stages, or germ cell counts [31]. It is possible that the MPs with SAR used in the current study and previous studies did not produce enough temperature effects, as several studies have demonstrated that the effects of MP exposure on the male reproductive system may be due to the cumulative effects of the heat generated [59,60,61]. In our study, we measured the temperature and found that, during EMR exposure, no temperature increase was observed, and no abnormalities in semen quality were noted.

One potential limitation of the current study is that only male beagles were used, since we also aimed to assess the effect of EMR on semen quality. We do not know whether or not EMR affects female reproduction. The lack of a control group, performed simultaneously with the experimental group, is another limitation. In the current study, we compared several timepoints during exposure to EMR with the baseline. In this case, the statistical design was set up to eliminate this weak point; therefore, the lack of a control group may not interfere with the outcomes of the study. In addition, the SAR used in the current study was in the general range of current MPs available in the market (i.e., 0.279–1.557 W/kg), and the duration of MP exposure in the study was limited to 10 consecutive weeks. The use of a higher SAR and longer duration of exposure may lead to adverse effects, which requires further investigation. Furthermore, the positions of MPs in the current study were on both sides of the chest. We are aware that the MP should be placed near the target organ that was evaluated (i.e., testes). The present study aimed to evaluate effects of MP on several systems (heart and circulation, hematology, and reproductive system); therefore, we decided to place the MPs on the chest. In addition, it is safer for the MP to be placed in the pocket of the dog’s jacket. Lastly, the small sample size used in the current study (n = 7) may affect the inference of the results; therefore, results should be interpreted with caution. Further study, with a larger sample size, will ensure the current results.

## 5. Conclusions

Exposure to EMR from MPs for 2 h daily, 5 days a week, for a total of 10 consecutive weeks, led to no adverse outcomes on the HRV, cardiovascular function, blood profiles, or semen quality in healthy conscious dogs. The slight reduction in HR, measured based on HRV and ECG, could be a random variation, rather than an effect of EMR exposure. The findings from this study need to be supported by further studies, with a more prolonged exposure time or incorporating more specific molecular parameters.

## Figures and Tables

**Table 1 vetsci-09-00201-t001:** Time- and frequency-domain parameters of heart rate variability (HRV) from seven dogs at baseline, weeks 2, 4, 6, 8, and 10.

Parameters	Baseline	Week 2	Week 4	Week 6	Week 8	Week 10
**Time-Domain Parameters**						
NNA (ms)	719.6 ± 88.6	726.8 ± 89.0	711.2 ± 98.4	782.9 ± 113.8 *	767.8 ± 115.7	749.3 ± 108.2
SDNN (ms)	278.0 ± 43.3	272.4 ± 57.0	262.0 ± 59.6	298.3 ± 71.2	281.2 ± 73.2	274.2 ± 55.2
SDANN (ms)	83.5 ± 25.7	89.1 ± 18.8	67.2 ± 7.7	93.1 ± 21.9	91.6 ± 22.2	82.4 ± 16.0
SDNN index (ms)	264.5 ± 45.0	256.2 ± 58.1	254.3 ± 62.7	282.7 ± 70.5	264.5 ± 72.7	260.8 ± 55.1
rMSSD (ms)	288.5 ± 97.3	285.5 ± 94.4	286.3 ± 101.2	327.1 ± 108.3	301.9 ± 108.7	313.3 ± 88.0
pNN50 (%)	72.2 ± 8.9	72.6 ± 7.1	73.8 ± 10.6	76.2 ± 8.8	75.1 ± 8.5	75.6 ± 10.2
**Frequency-Domain Parameters**						
ULF (ms^2^)	2732.5 ± 1067.1	2683.8 ± 795.7	2905.6 ± 1605.2	2688.8 ± 888.5	2605.0 ± 950.3	2649.8 ± 572.4
VLF (ms^2^)	9704.0 ± 3563.0	10,142.2 ± 3259.6	11,175.1 ± 6806.4	10,881.0 ± 5533.4	10,629.2 ± 5231.8	9498.3 ± 3162.1
LF (ms^2^)	7470.5 ± 1924.3	7381.5 ± 2889.7	10,802.6 ± 12,455.8	9789.5 ± 5190.5	8725.1 ± 5284.0	6725.1 ± 2395.6
HF (ms^2^)	46,195.1 ± 20,388.7	43,781.7 ± 24,415.9	45,621.2 ± 27,983.4	56,424.6 ± 33,653.3	50,087.8 ± 35,977.0	44,644.9 ± 22,207.1
TP (ms^2^)	66,102.0 ± 25,515.7	63,989.1 ± 30,693.2	70,504.6 ± 47,330.6	79,784.0 ± 44,745.7	72,047.1 ± 47,126.1	63,518.1 ± 27,644.1
LF/HF	0.18 ± 0.05	0.19 ± 0.05	0.21 ± 0.10	0.19 ± 0.04	0.19 ± 0.04	0.17 ± 0.04
LF nu	0.15 ± 0.04	0.16 ± 0.03	0.17 ± 0.06	0.16 ± 0.03	0.16 ± 0.03	0.14 ± 0.03
HF nu	0.85 ± 0.04	0.84 ± 0.03	0.83 ± 0.06	0.84 ± 0.03	0.84 ± 0.03	0.86 ± 0.03

All data are presented as mean ± SD. * Indicates *p* < 0.05, when compared with baseline, using repeated one-way ANOVA with Dunnett’s post hoc analysis. NNA: mean of the interval between continuous normal R waves, SDNN: the standard deviation of all NN-intervals, SDANN: the standard deviation of the averages of NN-intervals in all 5 min periods of the entire recording, SDNN index: mean of the standard deviation of the averages of NN-intervals in all 5 min periods of the entire recording, rMSSD: the root mean square of successful differences between adjacent NN-intervals, pNN50: the percentage of adjacent NN-intervals differing by more than 50 milliseconds in the whole recording divided by the total number of all NN-intervals, ULF: ultra-low frequency, VLF: very low frequency, LF: low frequency, HF: high frequency, TP: total power, LF nu: normalized LF power, and HF nu: normalized HF power.

**Table 2 vetsci-09-00201-t002:** Electrocardiography parameters obtained from seven dogs at baseline, week 5, and week 10 after exposure to EMR from MPs.

Parameters	Baseline	Week 5	Week 10
RR (ms)	462.2 ± 100.3	557.5 ± 106.2 ***	478.5 ± 60.0
PQ (ms)	84.0 ± 8.8	84.6 ± 8.3	85.5 ± 7.3
QRS (ms)	41.0 ± 2.7	41.4 ± 3.1	41.1 ± 2.2
QT (ms)	175.1 ± 9.5	191.1 ± 14.2 **	180.6 ± 10.7
QTc (ms)	221.9 ± 8.0	229.6 ± 13.1	226.0 ± 10.5

All data are presented as mean ± SD. ** Indicates *p* < 0.01, *** *p* < 0.001, when compared with baseline using repeated one-way ANOVA with Dunnett’s post hoc analysis. RR: the duration between two continuous R waves, PQ: the duration between the start of a P wave and QRS complex, QRS: the duration between the start of a Q-wave and the ending of an S wave, QT: the duration between the start of a Q-wave and ending of a T-wave, QTc: corrected QT (Van de Water).

**Table 3 vetsci-09-00201-t003:** Echocardiographic parameters from seven dogs at baseline, week 5, and week 10.

Parameters	Baseline	Week 5	Week 10
LA/Ao	1.28 ± 0.09	1.28 ± 0.07	1.33 ± 0.11
IVSd (cm)	0.86 ± 0.12	0.92 ± 0.08	0.97 ± 0.17
IVSs (cm)	1.08 ± 0.12	1.12 ± 0.15	1.26 ± 0.22
LVIDDN (cm)	1.41 ± 0.10	1.41 ± 0.15	1.38 ± 0.18
LVIDSN (cm)	0.84 ± 0.04	0.80 ± 0.10	0.78 ± 0.14
LVPWd (cm)	0.80 ± 0.08	0.85 ± 0.08	0.93 ± 0.15
LVPWs (cm)	1.15 ± 0.09	1.17 ± 0.13	1.20 ± 0.14
FS (%)	37.1 ± 3.9	39.7 ± 4.8	40.6 ± 5.0
HR (bpm)	117 ± 25.8	104 ± 22.7	122 ± 11.2
EDVI (mL/m^2^)	37.7 ± 7.9	37.1 ± 6.2	37.2 ± 6.3
ESVI (mL/m^2^)	9.3 ± 2.4	8.2 ± 2.7	8.8 ± 2.7
SV (mL)	15.6 ± 4.1	15.9 ± 2.9	16.0 ± 3.2
CO (L/min)	1.8 ± 0.6	1.6 ± 0.3	1.9 ± 0.4
EF (%)	75.3 ± 3.2	78.4 ± 3.8	76.5 ± 4.9
E/A	1.8 ± 0.3	1.9 ± 0.3	1.7 ± 0.3
AV PG max (mmHg)	6.4 ± 3.4	5.0 ± 2.8	6.6 ± 1.6
IVRT (ms)	45.3 ± 15.7	44.1 ± 7.8	40.9 ± 10.6
IVCT (ms)	36.7 ± 9.2	36.7 ± 6.6	33.9 ± 7.6
LVET (ms)	155.3 ± 20.9	155.6 ± 10.8	157.7 ± 18.5
Tei index	0.54 ± 0.22	0.52 ± 0.07	0.47 ± 0.09

All data are presented as mean ± SD. LA/Ao: left atrium to aortic root ratio, IVSd: interventricular septum diastole, IVSs: interventricular septum systole, LVIDDN: normalized left ventricular internal diameter in diastole, LVIDSN: normalized left ventricular internal diameter in systole, LVPWd: left ventricular posterior wall diastole, LVPWs: left ventricular posterior wall systole, FS: fractional shortening, HR: heart rate, bpm: beats per minute, EDVI: left ventricular end-diastolic volume indexed to body surface area, ESVI: left ventricular end-systolic volume indexed to body surface area, SV: stroke volume, CO: cardiac output, EF: ejection fraction, E/A: left ventricular early (E) to late (A) ventricular filling velocities ratio, AV PG max: maximum of aortic valve pressure gradient, IVRT: isovolumetric contraction time, IVCT: isovolumetric relaxation time, and LVET: left ventricular ejection time.

**Table 4 vetsci-09-00201-t004:** The complete blood count (CBC) and plasma chemistry profiles obtained from seven dogs at baseline, week 5, and week 10 after exposure to EMR from MP.

Parameters	Baseline	Week 5	Week 10	Normal Range
**Complete Blood Count**				
RBC (10^6^/µL)	6.6 ± 0.4	6.5 ± 0.6	6.8 ± 0.7	5.1–8.5
Hemoglobin (g/dL)	15.4 ± 1.0	15.3 ± 1.3	15.4 ± 1.6	11.0–19.0
Hematocrit (%)	41.5 ± 2.8	41.4 ± 3.3	43.6 ± 4.4	33.0–56.0
MCV (fL)	62.5 ± 2.4	64.2 ± 1.6	64.7 ± 1.1	60.0–76.0
MCH (pg)	23.2 ± 0.7	23.7 ± 0.6	22.9 ± 1.1	20.0–27.0
MCHC (g/dL)	37.1 ± 1.0	37.0 ± 0.8	35.4 ± 1.4 *	30.0–38.0
Platelets (10^3^/µL)	227.4 ± 42.4	173.7 ± 102.4	227.7 ± 43.6	117.0–490.0
WBC (10^3^/µL)	11.0 ± 1.9	10.9 ± 3.0	7.3 ± 1.3 ***	6.0–17.0
Neutrophils (10^3^/µL)	6.0 ± 0.9	5.6 ± 1.3	3.8 ± 0.8 ***	3.62–12.30
Lymphocytes (10^3^/µL)	3.5 ± 1.3	3.5 ± 1.7	2.7 ± 0.9	0.83–4.91
Monocytes (10^3^/µL)	1.0 ± 0.3	1.2 ± 0.4	0.7 ± 0.1	0.14–1.97
Eosinophils (10^3^/µL)	0.48 ± 0.13	0.51 ± 0.22	0.11 ± 0.04 ***	0.04–1.62
Basophils (10^3^/µL)	0.041 ± 0.025	0.029 ± 0.019	0.004 ± 0.005 ***	0.00–0.12
**Plasma Chemistry**				
ALT (U/L)	49.6 ± 11.1	50.4 ± 7.9	46.6 ± 8.6	4.0–91.0
AST (U/L)	22.4 ± 3.2	28.0 ± 4.9 **	33.9 ± 7.9 **	10.0–59.0
ALP (U/L)	92.4 ± 37.7	93.6 ± 48.0	92.1 ± 44.5	3.0–61.0
BUN (mg/dL)	10.5 ± 3.9	18.9 ± 3.8 ***	18.9 ± 5.4 **	7.0–30.0
Creatinine (mg/dL)	0.6 ± 0.1	0.5 ± 0.2	0.6 ± 0.2	0.6–2.0
Total protein (g/dL)	5.2 ± 0.1	5.5 ± 0.3 *	5.8 ± 0.3 ***	5.8–8.8
Albumin (g/dL)	2.34 ± 0.08	2.44 ± 0.15	2.59 ± 0.12 ***	2.6–4.3

All data are presented as mean ± SD. * Indicates *p* < 0.05, ** *p* < 0.01, *** *p* < 0.001, when compared with baseline using repeated one-way ANOVA with Dunnett’s post hoc analysis. RBC: red blood cell, MCV: mean corpuscular volume, MCH: mean corpuscular hemoglobin, MCHC: mean corpuscular hemoglobin concentration, WBC: white blood cell, ALT: alanine transaminase, AST: aspartate aminotransferase, ALP: alkaline phosphate, and BUN: blood urea nitrogen. Normal ranges are from the Small Animal Teaching Hospital, Faculty of Veterinary Science, Chulalongkorn University.

**Table 5 vetsci-09-00201-t005:** The body surface temperature on the skin under the location of the mobile phone (MP) obtained from seven dogs at baseline, week 5, and week 10 after exposure to EMR from MPs.

Parameters	Baseline	Week 5	Week 10
Morning temperature (°F)	97.22 ± 0.82	97.54 ± 0.12	97.50 ± 0.27
Afternoon temperature (°F)	97.91 ± 0.52	97.86 ± 0.41	97.79 ± 0.31
∆ temperature (°F)	0.69 ± 0.64	0.33 ± 0.47	0.29 ± 0.25

All data are presented as mean ± SD. ∆ temperature: the changes in temperature throughout the day.

## Data Availability

The data presented in this study are available within the article.

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
