# Peer review of "The Preliminary Chronic Effects of Electromagnetic Radiation from Mobile Phones on Heart Rate Variability, Cardiac Function, Blood Profiles, and Semen Quality in Healthy Dogs"

_vetsci, 2022, doi:10.3390/vetsci9050201_

Round 1
Reviewer 1 Report
This is a very interesting study evaluating the effects of electromagnetic radiation exposure for 10 weeks on ECG, blood pressure, echocardiography, cTnI levels, hematology, biochemistry profiles, body surface temperature, and semen quality in 7 healthy dogs.
The study design is appropriate, the materials and methods are described in a clear and detailed manner, the results are valid, the discussion and the conclusions are supported by the results.
However, I would like to point out some minor revisions:
I think it is more appropriate to indicate the variability of the mean of your sample with the standard deviation rather than with the standard error. Please replace the standard error of the mean with the standard deviation of the mean in all values (tables and text) or to provide valid reasons for the use of the standard error instead of the standard deviation.
Again:
Lines 105-106: please indicate the nation in which the study has been carried out
Line 177: please replace the word ECC with ECG
Lines 553-564: the authors should indicate that the small sample size represents a limit for the inference of results.
Author Response
I think it is more appropriate to indicate the variability of the mean of your sample with the standard deviation rather than with the standard error. Please replace the standard error of the mean with the standard deviation of the mean in all values (tables and text) or to provide valid reasons for the use of the standard error instead of the standard deviation.
Response: Thank you reviewer, the SEM in all tables and results of the revised manuscript has been changed to SD.
Again:
Lines 105-106: please indicate the nation in which the study has been carried out
Response: Thank you reviewer, the sentence has been revised and now it is: “The study was conducted at the Chulalongkorn University Laboratory Animal Center (CULAC), Thailand and approved by the Institutional Animal Care and Use Committee of CULAC (Animal Use Protocol Number: 2073013).”
Line 177: please replace the word ECC with ECG
Response: Thank you reviewer, the typo has been revised.
Lines 553-564: the authors should indicate that the small sample size represents a limit for the inference of results.
Response: Thank you reviewer, the small sample size has been acknowledged in the potential limitation section as follow: “Lastly, the small sample size used in the current study (n=7) may affect the inference of the results; therefore, results should be interpreted with caution. Further study with more sample size will ensure the current results.”
Reviewer 2 Report
Dear Authors,
I found it very interesting to review a paper entitled "The chronic effects of electromagnetic radiation from mobile phones on heart rate variability, cardiac function, blood profiles, and semen quality in healthy dogs", although I have few comments.
The Authors state that EMR may impact reproductive functions in males as they carry MPs in their pant pockets (lines 83-85). In the light of that thought it would be much more appropriate if in tested animals the MPs would be placed close to the testicles and not on the chest. In the presented model the localization of the MPs should be mentioned as a limitation.
Another limitation of the study is the number of animals - only seven dogs were used what makes the results rather preliminary than certain. As the authors mention that the obtained results should serve as a base for further investigation, it should be also mentioned in the title of the paper.
The citation of references should be evaluated and corrected where applicable e.g. line 543.
Author Response
The Authors state that EMR may impact reproductive functions in males as they carry MPs in their pant pockets (lines 83-85). In the light of that thought it would be much more appropriate if in tested animals the MPs would be placed close to the testicles and not on the chest. In the presented model the localization of the MPs should be mentioned as a limitation.
Response: Thank you reviewer, we agree with the reviewer’s comment. The limitation about location of MPs was added to the limitation section as follow: “Furthermore, the positions of MP in the current study were on both sides of the chest. We are aware that the MP should be placed near the target organ that was evaluated (i.e., testes). The present study aimed to evaluate effects of MP on several systems (heart and circulation, hematology, and reproductive system); therefore, we decided to place the MP on the chest. In addition, it is safer for the MPs to be placed in the pocket of dog’s jacket.”
Another limitation of the study is the number of animals - only seven dogs were used what makes the results rather preliminary than certain. As the authors mention that the obtained results should serve as a base for further investigation, it should be also mentioned in the title of the paper.
Response: Thank you reviewer, the title of the paper was revised to “The preliminary chronic effects of electromagnetic radiation from mobile phones on heart rate variability, cardiac function, blood profiles, and semen quality in healthy dogs”
The citation of references should be evaluated and corrected where applicable e.g. line 543.
Response: Thank you reviewer, we have double checked and evaluated on the citation of references. We have added some more information into that sentence so that they are corrected for both information and style of the journal.
Reviewer 3 Report
Dear Authors,
the manuscript is interesting and well written, but the English language needs of a moderate revision.
Materials and methods need to be better detailed, in particularly procedures relating the semen collection and analyisis.
The important limitation of study is related to the low number of dogs considered that propably justify the poor significance of results. It may be interesting to evaluate data in dogs not included in experimental design.
Author Response
The manuscript is interesting and well written, but the English language needs of a moderate revision.
Response: Thank you reviewer, the authors are aware that we are not native English speaker. The manuscript has been submitted for English edited by “MDPI” and the certification was attached to this response.
Materials and methods need to be better detailed, in particularly procedures relating the semen collection and analysis.
Response: Thank you reviewer, detail of the semen collection and analysis have been added and highlighted in yellow color. We have copied those sections and put here so that reviewer can evaluate the changes. …..“Briefly, the semen ejaculation was collected into 15 mL Falcon conical tubes via digital manipulation with a latex glove. The method was performed in all dogs without the presence of an estrous bitch. Basically, the dog’s penis was vigorously massaged until the presence of bulbus glandis engorgement. Then the collector retracts the prepuce caudally past the bulbus glandis and applies firm constant pressure to the penis behind the bulbus glandis using index finger and thumb. The ejaculation is usually started after application of pressure behind the bulbus glandis. The total duration of induced ejaculation was 5 minutes per dog. The total semen volume was evaluated from three consecutive fractions which included the pre-sperm fraction, sperm-rich fraction, and prostatic fraction. Semen samples were kept at 37 °C until analysis.
The volume of the sample was determined using a graduate micropipette (Gilson, Gilson International France).
The sperm concentration was determined using a hemocytometer. A raw semen was placed in a channel for sperm counting and covered with its special coverslip to allow a very specific amount of fluid to be contained under the coverslip. The total sperm concentration was counted at 400× magnification using a phase contrast microscope (Olympus, Spain).
Morphological assessment of the sperm tail and head was performed using wet-mounting of formal saline-fixed samples and Williams staining, respectively. A total of 200 sperm cells were evaluated at 400× magnification. The percentages of normal head and tail morphologies were reported.
The percentage of sperm motility was evaluated immediately after collection. A 10 μL sperm sample was placed on a pre-warmed slide (37 °C) and covered with a pre-warmed coverslip. The motility was evaluated as total motility (i.e., the number of sperm moving). Initially, it was assessed at low power and finally and the final observations were made at 100× magnification using a phase contrast microscope (Olympus, Spain).
The important limitation of study is related to the low number of dogs considered that probably justify the poor significance of results. It may be interesting to evaluate data in dogs not included in experimental design.
Response: Thank you reviewer, we agree with the reviewer that the low animal number may affect the outcome of the study and we have added this limitation into the potential limitation section as follow: “Lastly, the small sample size used in the current study (n=7) may affect the inference of the results; therefore, results should be interpreted with caution. Further study with more sample size will ensure the current results.”

Round 2
Reviewer 3 Report
Well done.